# Current Practice and Potential Associated with Timber-Based Solutions for Buildings Retrofitting

Cláudio Meireis [1], Filipa S. Serino [1], Carlos Maia [1], André C. Fontes [1] and Jorge M. Branco [2,*]

[1] Lab2PT, School of Architecture, Art and Design, University of Minho, 4800-58 Guimarães, Portugal; a81783@alunos.uminho.pt (C.M.); a80896@alunos.uminho.pt (F.S.S.); cmaia@eaad.uminho.pt (C.M.); afontes@eaad.uminho.pt (A.C.F.)
[2] ISISE, Department of Civil Engineering, University of Minho, 4800-058 Guimarães, Portugal
* Correspondence: jbranco@civil.uminho.pt

**Abstract:** Current buildings are responsible for the highest energy consumption, exceeding polluting sectors such as industry and transports. In Portugal, a large part of the building stock was built in the 1970s and 1980s, but buildings dated from the 1960s and 1970s are the ones with the most anomalies and worst quality of construction and, therefore, worst energy performance. The renovation of those buildings can represent an excellent opportunity to correct and improve their energy deficiency and, with that, to promote a more sustainable building stock. The ETICS system is the most used for the renovation of buildings in Portugal due to its lower cost, quick application and thermal efficiency, but it doesn't solve other problems that may exist, such as structural safety and interior organization of the existing building. The application of prefabricated systems in the envelope has proved to be successful in improving energy efficiency, allowing new volumes and extra areas while contributing to the structural resilience of existing buildings. This paper aims to describe the current situation of the buildings renovation in Portugal and to discuss the potential of innovative envelope retrofitting solutions, using natural materials like timber, and is more concerned with the problems of existing buildings and the need for comfort and space for the occupants.

**Keywords:** building stock; retrofitting; energy efficiency; architectural transformation; structural resilience; timber



## 1. Introduction

According to "Instituto Nacional de Estatística" [1], in 2019, the Portuguese building stock was composed of almost 3.6 million buildings and 5.9 million dwellings. About 63.1% of the building stock was built in the 1970s and 1980s, but buildings dated from the 1960s and 1970s are the ones showing more pathologies with less comfort and lower energy performance. In developed countries, the energy consumption in residential and commercial buildings is between 20% and 40% of the total energy consumption, exceeding the sectors of industry and transport [2]. In the case of Portugal, both high energy consumption and significant energy losses through the envelope are consequences of the lack of regulation and thermal requirements when buildings were built. Studies show that, to reduce buildings' energy consumption, it is necessary to act mainly in its envelope, where the energy losses are higher. For example, the addition of thermal insulation can lead to reductions in heating needs by up to 70% [3,4]. In practice, ETICS (External Thermal Insulation Composite System) is the most common retrofitting system used in Portugal; however, although it has a high energy efficiency and quick and simple application, it doesn't solve other problems that may exist. The cost-benefit assessments of retrofit actions in this sector show excessive payback times, creating a strong and generalized lack of confidence by investors and final users. Regarding the energy retrofit, [5] showed that it is possible to overcome the barriers of high costs and long technological solutions using prefabricated systems for low energy renovation. Most of these solutions are generally founded on the

load bearing capacity of the existing buildings, a condition that is rarely applicable in the highly seismic areas of Mediterranean countries. Thus, in many EU regions, it became imperative to couple energy retrofit with structural safety improvement, to make visible that higher initial investments of retrofit are more interesting in the long term than lower investments with higher paybacks. On the other hand, the confinement derived from the present pandemic caused by the SARS-CoV-2 virus has highlighted the need to incorporate new functions and uses into housing. The post-pandemic housing model should incorporate flexible and new programmatic areas: for teleworking, differentiated spaces for video conferencing, leisure separated from the living areas, outdoor areas, garden, etc. In this regard, the strategies for the renovation of existing buildings should improve three main areas: energy efficiency, structural stability and architectural renovation.

## 2. Current Practice for Buildings Renovation in Portugal

In Portugal, the current strategy to reduce energy consumption consists of the improvement or replacement of existing window frames (high-performing windows) and in the application of thermal insulation on walls and roofs. The most common thermal insulation solution is the ETICS system (External Thermal Insulation Composite System) composed of a thermal insulation board, generally EPS (Expanded Polystyrene insulation), with a thin plaster reinforced with fiberglass or synthetic mesh and an appropriate coating which can be painted or coated to give a traditional appearance [6]. This system became popular due to its high thermal efficiency, the correction of thermal bridges that can be achieved and consequent reduction of the risk of condensation, and the preservation of thermal inertia since it is placed on the external façade, while representing a low investment cost. Moreover, the quick and simple application in façades with few architectural features, and the convenience of the occupants during the execution (the inner area remains intact), are the features that contributed to the dissemination of this technique. However, the ETICS system does not allow the aesthetical preservation of buildings with architectural or heritage value, the edge finishing execution is complex due to architectural constraints, it has vulnerability to mechanical stresses (impacts) and no fire resistance due to the combustibility of the EPS insulation board, the execution exposed to weather conditions may compromise the performance of the thermal retrofitting solution, and it is not appropriate for façades with ascending humidity or façades composed of very porous materials [7]. Moreover, the appearance of anomalies and durability problems with the system's coating is common, such as material rupture anomalies (oriented cracking, non-oriented cracking, deterioration of the covering of reinforcement, detachment of the finishing coat, partial or total loss of adherence and material gap), color/aesthetic anomalies (efflorescence, runoff marks, corrosion stains, graffiti, biological growth and other color changes) and flatness anomalies (flatness deficiency, surface irregularities, joints between plates visibility, swelling of the finishing coat and swelling of the insulation plates) [8].

In addition to the ETICS system, solutions like "thermal insulation injection in the cavity of double-leaf walls", "internal thermal insulation" and "mortar with improved thermal performance" are also used but in more specific cases. The thermal insulation injection in the cavity of double-leaf walls can be made through the injection of insulation products in granules or foam (expanded on-site). This solution is mainly used when the façade is composed by double-leaf walls, and it is intended to preserve the original internal and external appearance of the façades. However, this solution does not protect the façade from outside actions caused by atmospheric agents, does not eliminate thermal bridges, the execution might have some complexity depending on the façade features and the efficiency of its implementation depends on the existing conditions in the cavity. The internal thermal insulation solution can be made through the ITICS system (Internal Thermal Insulation Composite System) or counter-wall of light brick masonry or gypsum plasterboards. This solution can be applied when the façade is composed of a single leaf wall, and it is intended to preserve the appearance of old buildings with architectural or heritage value. However, with this solution, there are some disadvantages, such as the reduction of the inner area,

the constraints for the occupants during its execution, the elimination of thermal inertia of the existing wall and the permanence of thermal bridges. The mortar with improved thermal performance solution can be applied in the external or internal cladding of the façade. Although this solution increases the acoustic properties of the façades, and has easier execution and permeability to water vapor, it also has less thermal efficiency when compared with other solutions [7].

## 3. Innovative and Integrated Solutions

In the last years, some research and case studies, in particular, within European projects, have been addressing the development and the promotion of retrofitting solutions able to integrate more than just the thermal insulation. Attempts have been made to incorporate architectural and structural demands. This section aims to summarize the ones that have presented higher potential.

### 3.1. Prefabricated Cladding Solutions

The application of prefabricated claddings in the envelope of existing buildings has been addressed in recent European projects, such as the RetroKit and the BRESAER. Within the Retrokit project [9], a multifunctional façade based on prefabricated modules used on the façade and roof has been developed. These modules integrate solutions that deal with the aspects of heating, ventilation, cooling and electricity in a flexible way. The overall life cycle cost is reduced due to easy installation, simple maintenance since the pipes on façades are easily accessible from the outside, and convenient recycling. Since the prefabricated modules are applied on the exterior, it is possible to solve the problem with the thermal bridges, corners and connections. Moreover, this system improves the aesthetics, comfort, energy performance and property value of the existing building.

The European Project BRESAER [10] developed a prefabricated retrofitting system adaptable to different climate zones through a variable choice of highly efficient energy technologies and intelligent controls. This system is composed of a series of structural and technological envelope elements that are placed on the outer side of the façade and roof and can be adapted to the particular needs of each project. Besides the insulation layer and the support structure, this system consists of solar thermal air heating panels that can be placed on façades and roofs, ventilated façade panels with a lightweight concrete exterior, multifunctional insulating panels with lightweight concrete support, dynamic window replacement modules with shading control, and photovoltaic panels that can be placed on envelope elements (Figure 1).

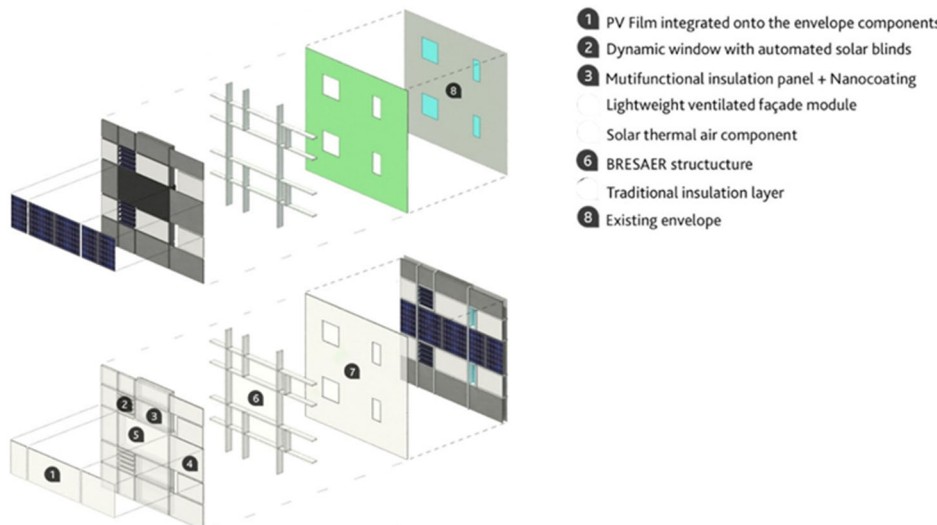

**Figure 1.** Schematic representation of the BRESAER system for façades [10].

The combination of active and passive prefabricated solutions allows the system to reduce the primary energy and greenhouse emissions while improving indoor environment quality. Currently, the passive solutions used are the sun protection devices for windows, Trombe walls and the maximization of natural ventilation. An example of passive solutions is the ventilated façade retrofitting system E2VENT [11] that, besides integrating thermal insulation, also introduces PCM (Phase Change Materials), which, when it changes the physical state, absorbs or releases heat, allowing it to passively control the room temperature and reduce 50% of primary energy needs. Although these systems allow multifunctional interventions and adaptability to different climate zones, they don't take full advantage of the existing buildings, such as spatial organization. In other words, it does not qualify the pre-existing interior spaces, maintaining a static and rigid organization.

### 3.2. Double Skin and Exoskeleton Solutions

The double-skin façade consists of an external glazing, an intermediate thermal buffer zone and an inner façade. With this solution, it is possible to reduce the energy consumption and enhance the thermal comfort through the renovation of the building's envelope and combination of different typologies and adjustable systems, such as climate conditions, ventilation and depth. Moreover, with the addition of a shading system, inside or outside the double-skin façade, adjustable according to the sunlight angles, the glare problems can be avoided, and the natural lighting can be uniformly distributed inside the existing building. The acoustic insulation is also guaranteed, since the outer skin works as a barrier against the noise, obstructing its propagation. In this regard, the introduction of a double-skin façade solution for the renovation of existing buildings allows the enhancement of thermal comfort, control of daylighting and glare, sound insulation, noise mitigation and structure stability [12].

The exoskeleton system consists of a structure that helps to support other elements. This system is generally used to support technical rooms and mechanical systems; however, its adaptation to the renovation of the envelope can result in the extension of the internal space of the building, creating greater flexibility and adaptability to the needs of people, as well as greater structural safety and energy efficiency. The main difference between double-skin façade and exoskeleton is its support. The exoskeleton solution is supported through its foundation, while the double-skin solution is supported by the existing structure, requiring no foundations. In general, in most of the existing buildings, infill walls are one of the most scattering sources in terms of heat loss. Therefore, ref. [13] proposed a double-skin solution with new infilled RC frames externally added and connected to the existing RC structure to satisfy both thermal and seismic requirements (Figure 2). With this solution, it was possible to contribute to the lateral load bearing capacity in terms of strength and stiffness and to reduce the energy consumption. Reference [14] also proposed an external integrated double-skin façade solution to improve the architectural and urban environment quality, the energy efficiency and the structural performance, regarding the minimum environmental impact principles, the minimum rehabilitation cost requirement and the minimum impairment of the occupants. This solution was conceived as an exoskeleton with a double value: on one hand, the structure provides the existing buildings with the necessary seismic resistance and its dry installation does not require prolonged phases of construction, and on the other hand, the external solution guarantees the minimum impact on the occupants during the building rehabilitation and allows future functional and formal variations. Moreover, the enlargement of the existing building structure allows for the creation of new areas, such as new living spaces, balconies and solar greenhouses.

The improvements in terms of energy performance and structural safety of the intervention have been investigated and have proved the high potential of the expandable architecture in a cost-effective analysis. For example, in an ongoing European Project, Pro-GET-onE, an exoskeleton made of a steel frame (two columns and a beam) for each floor with bracings in the transversal direction connected to the column-beam joints of the existing building, has been proposed. The design of adaptable typologies was also

considered through different solutions with different façade and room combinations, such as balconies, extra rooms and sunspaces (Figure 3). However, the disadvantages that this system may have are related to the reduction of daylight due to the depth of the system's structure [15].

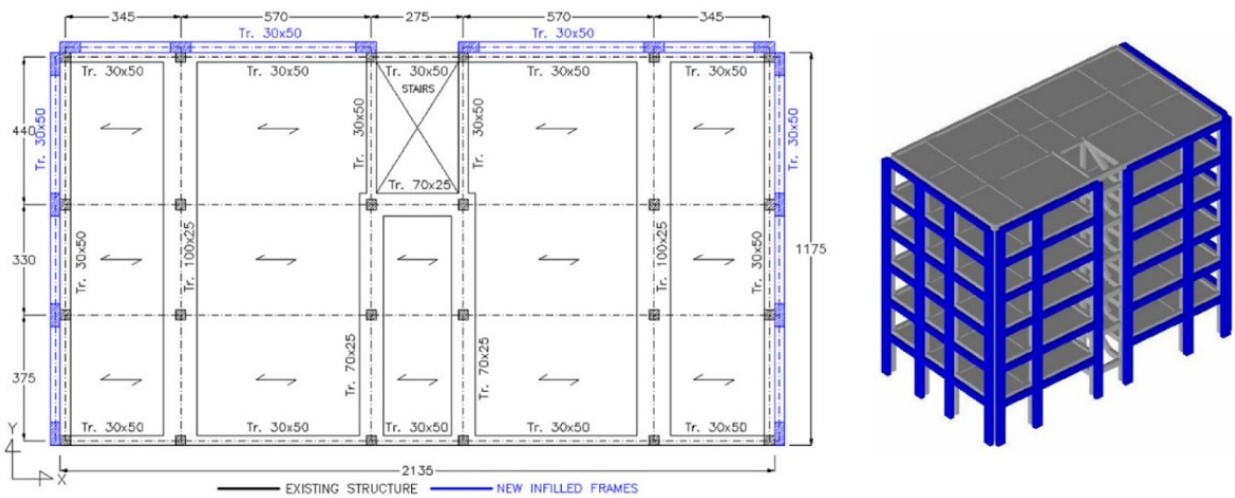

**Figure 2.** Double-skin solution with new infilled RC frames externally added and connected to the existing RC structure (dimensions in centimeters) [13].

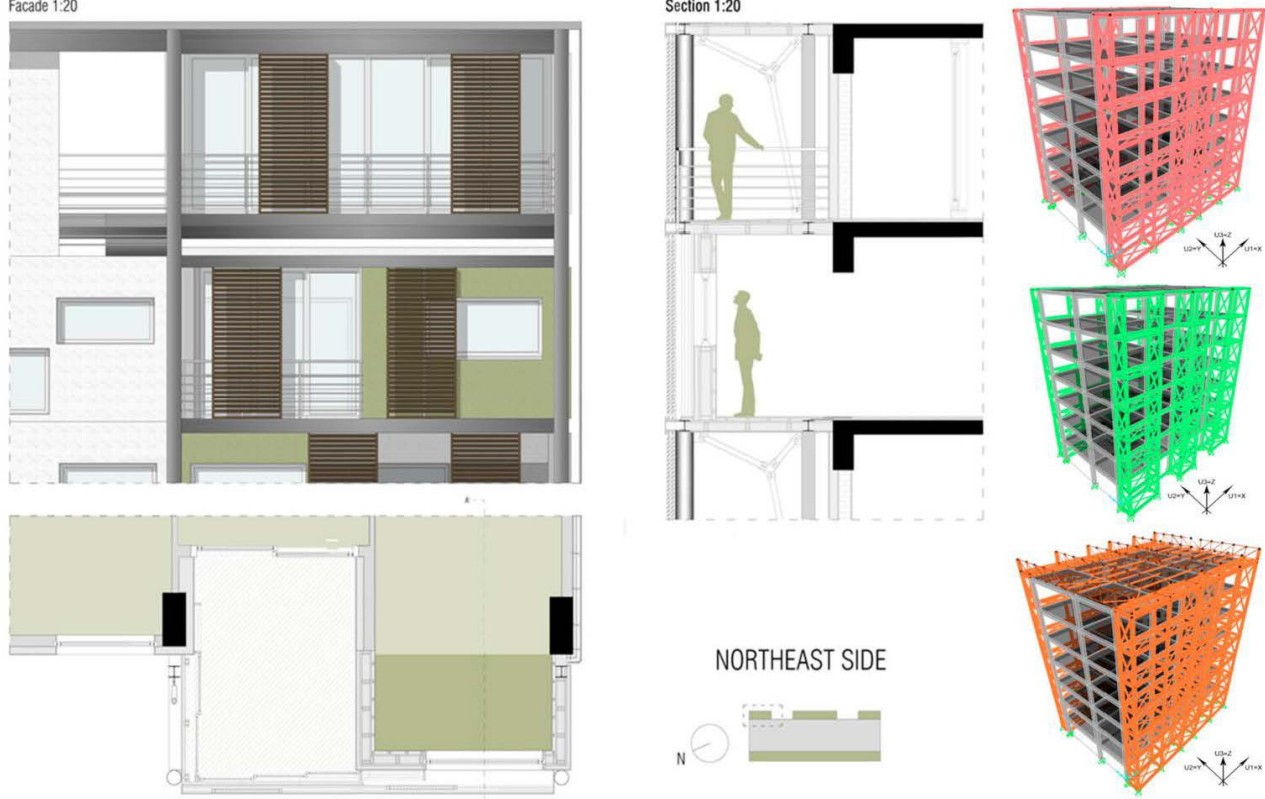

**Figure 3.** Schematic representation of the Pro-GET-onE system [15].

The use of the exoskeleton allows the system to obtain several architectural solutions, such as the "Five morpho-techno-typological solutions" proposed by [16], which consist of five typologies that can be adapted to the needs of people and the building, and the strategies on interventions on residential heritage proposed by [17], which consist of

intervention concepts that can be adopted depending on the architectural ideas and the level of degradation of the building.

An excellent example of the architectural transformation allowed by the exoskeleton solution is the project "Transformation de 530 logements" [18], by Anne Lacaton e Jean Philippe Vassal, that consists of the transformation of three inhabited social buildings. In this project, a metallic exoskeleton was used to extend the interior space and improve the energy efficiency and the quality of the building. With these prefabricated modules, it was possible to add new areas, such as winter gardens and balconies, and redesign the façades of the existing building. The general economy of the project was based on the choice of conserving the existing building without making important interventions on the structure, the stairs and the floors. Furthermore, the project dealt with the global performance of the building envelope, the reconfiguration of vertical circulations and access halls. The existing windows and sills were removed and the concrete walls were opened to introduce the interior thermal curtain and the aluminum frame sliding doors with access to the spaces created. This intervention made it possible to increase the living area, with the addition of the winter garden and balconies, and improve the energy performance of the building by 60%. Although the total cost of the project was lower than the estimated cost of the demolition and reconstruction, it remains relatively high [19]. The volumetric increase of the existing buildings is the main limitation for the exoskeleton-based integrated systems; indeed, it is not always achievable due to lack of available perimetral space and legislative constraints.

*3.3. Wood Based Solutions*

With the development of new engineered wood products (EWP), such as glulam and the more recent cross-laminated timber (CLT) panels and laminated veneer lumber (LVL), new possibilities for more sustainable interventions and prefabricated solutions have emerged [20]. Those EWP are improved products when compared with solid wood in terms of physical and mechanical performances, while sharing the same sustainable profile, being made of natural and renewable resources with the ability to store CO2. Moreover, the use of EWP promotes a better knowledge and control on the natural durability, preservation and treatment of the mass timber elements, crucial to develop maintenance plans [21].

There are some projects and systems that explore the energy renovation of existing buildings through the application of timber elements for the structure or cladding. The wood envelope solutions from "Rubner Holzbau" consist of prefabricated wall elements that work as a façade cladding system. In addition to the heat and noise comfort, and fire protection, the prefabrication of wooden elements allows the reduction of construction time as well as cost optimization and quick assembly [22]. The European Projects BERTIM and MORE-CONNECT are examples of systems that explore the renovation of existing buildings through the application of wood prefabricated modules on the façade. The BERTIM system is characterized by the integrated renovation process based on customized mass manufacturing methodologies supported by building information models (BIM). This tool enables the reduction of renovation operation time and makes the renovation process more efficient, through the customization of the mass production, from data gathering, designing, manufacturing and installation [23]. The MORE-CONNECT project focuses on the development of cost-optimal deep renovation solutions towards nearly zero energy buildings (nZEB). To achieve this goal, the introduction of natural materials, such as timber elements, represents an important role since it works as a carbon sponge due to the ability of store carbon dioxide, it is a natural and renewable resource, and it has low embodied energy (total energy used in the building construction process, from production to transportation of materials) [24]. The use of prefabricated panels not only allows the reduction of energy losses through the façade but also eases the maintenance of the buildings. The iNSPiRe project [25] presents an idea of minimal renovation developed through systemic renovation packages that can reduce the primary energy consumption of a building for ventilation, heating/cooling, domestic hot water and lightning. To support

the technologies and mechanical systems, a timber frame structure was used. These are some examples of the introduction of wood-based materials for the energy renovation of the existing buildings. However, the use of mass timber solutions is a better option since it increases the energy efficiency, structural safety and architectural renovation of the existing buildings. In this regard, [26] proposed an innovative energy, seismic, and architectural renovation solution for reinforced concrete (RC) framed buildings, based on the addition of CLT panels to the outer walls, in combination with wooden-framed panels. This system is based on the idea of cladding existing RC framed buildings with a new performing skin made of prefabricated and customizable elements. The CLT is an EWP with high strength, stiffness and dimensional stability. Therefore, the CLT structural panels are combined with non-structural pre-assembled panels, which are provided with high-performing windows that will replace the existing ones. This solution uses bio-based insulation materials, such as hemp, cork, wood fiber and cellulose fiber, and in the finishing layer can be used ceramic, wood, stone, glass, photovoltaic modules, etc. The support of the two panels is made through steel profiles connected to the existing RC structure, allowing a quick and easy external installation, which are adaptable to the most common RC framed buildings. With this solution it is possible to reduce the global energy consumption to nearly 60%. The European Project Pro-GET-onE also proposes an exoskeleton made of post-tensioned timber frames and other prefabricated façade elements that increase the energy efficiency and structural safety of the existing buildings (Figure 4). The construction and application of this system can be divided in two parts: firstly, the post-tensioned timber frame is built, and secondly, the insulating façade elements are attached to the structure [27]. The use of CLT, wood-based systems or exoskeletons are good solutions to improve the current environmental and energy efficiency problems and respond to the current renovation requirements of quick installation, cost-effectiveness, use of low-carbon materials and reversibility. However, there are not many studies and systems about integrated wood-based retrofitting renovations; therefore, the potential of these solutions must be further investigated.

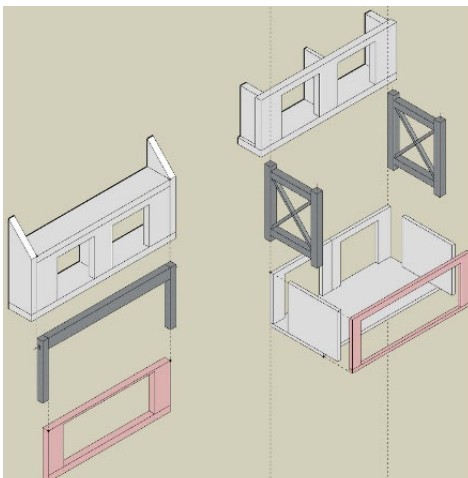

**Figure 4.** 2D and 3D timber frame systems developed within Pro-GET-onE [27].

## 4. Discussion

Past design studies and the performed technical–economical evaluation demonstrate that energy efficiency in residential urban complexes can be considered as an extraordinary opportunity to restore environmental, social and urban quality [28]. For example, the possibility to combine add-ons on the top and sides of the buildings (and thus new residential units) with facade solutions have been explored. The need to combine social, safety and energy issues is clear, integrating technical and architectural feasibility of zero energy and

low carbon areas with the creation of self-controlled safe environments in emerging new forms of urban communities.

The authors believe that the answer to these new requests could be found in what we call programmatic intermediate zones that can be added to the pre-existing buildings envelope, composed of wood-based systems. In a same holistic and integrated system based on timber pre-assembled components, we should combine the highest performances in terms of energy requirements, safety, social sustainability and market attractiveness. This goal is attained through the application of timber-based solutions for the building envelopes, as well as through an optimum climatic-structural-functional management, grounding on the substantial increase of the real estate value of the buildings through significant energy and architectural transformation. This incremented value will be the result of the development and application of integrated efficient technologies on existing buildings, providing the increase of structural safeness and energy efficiency. The proposal promotes the highest transformation of the existing building's shell with external added volumes, which generate energy efficient buffer zones and, at the same time, increase the building's volume (with balconies, sunspaces and extra rooms). It is carried out from the outside, therefore avoiding building downtime and inhabitant relocation, and it targets eco-efficiency by reducing the environmental impact of the renovation system from the construction to the end-of-use.

The structure of the prefabricated system is composed of glulam columns and beams and CLT slabs. The columns allow the continuity of spaces, and, since there are buildings with an irregular exterior envelope, generally caused by balconies, the use of CLT panels for the slabs will allow it to adapt to any kind and geometry of construction. In fact, in factory, and using the available digital technology of drawing and CNC cuts, it is possible to have a geometry of the CLT panel fitted to the existing building. As wood is a light material (strength/weight ratio), its use in this system allows the absence of additional foundations, with the prefabricated modules being directly supported by the reinforced concrete structure of the existing building.

In Figure 5, the components of the proposed system are represented in axonometry through the decomposition of elements. In addition to identifying the layer and their functions, it is also possible to perceive the relationship between the different elements.

Regarding thermal insulation, mineral wool was chosen to guarantee greater safety against fire and greater thermal and acoustic comfort. As wood has a low thermal conductivity, the insulation is placed in the interstices of the structural layer. Thermal bridges are minimized through the thickness of the thermal insulation and the placement of a wood fiberboard, which acts as additional insulation.

The interior cladding is made of OSB (Oriented Strand Board) panels and the exterior cladding is composed of wooden slat with a surface treatment and protection. The choice of wooden slats as an exterior coating is based on its versatility, since, depending on the distance between the vertical elements, it guarantees privacy for occupants and it controls the sunlight. The wooden substructure, with vertical and horizontal slats, not only supports the exterior cladding but also allows for the creation of a ventilated façade.

Although there are various functions that the building envelope must fulfill, there are layers that can perform several functions at the same time. In this regard, the structure functions as a structural layer and interior cladding; the OSB panel has the function of interior cladding, but also performs the function of airtightness through the placement of single-sided adhesive tapes, and a vapor barrier; the wood fiberboard is pressure resistant and vapor permeable and has the function of additional insulation, protection of structural components, and also a second layer resistant to water and wind.

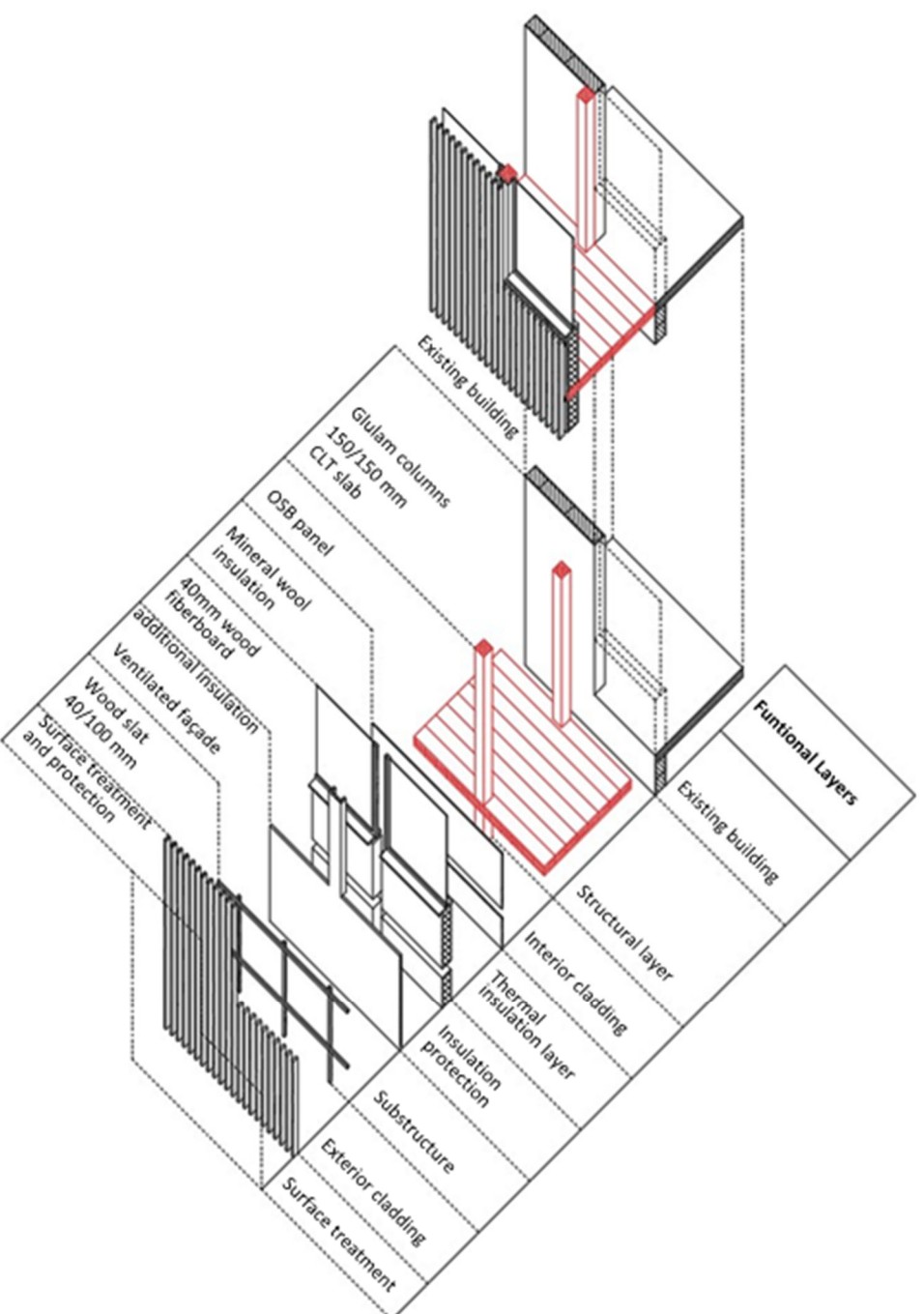

**Figure 5.** Structural and functional layers of the proposed prefabricated system.

One of the properties of wood is its combustibility. This is one of the main reasons why people, in general, do not feel safe in buildings with wooden structures, particularly tall buildings [21]. Although wood is a combustible material, there are two distinct approaches that allow a sufficient fire performance of a solid timber structural system: charring and encapsulation. Charring is a process in which, when exposed to fire, the outer layer of wood reaches its point of combustion, creating a layer of charcoal. This outer layer has a low conductivity, reducing fire progression. The inner core of the wood retains its mechanical properties, and only its moisture content is reduced. Therefore, charring is an approach that allows exposing the solid timber structure outside the building, while encapsulation is an approach where the system is coated with fire-resistant materials such as plasterboard panels. In this regard, depending on the height of the buildings in which it

will be intervened, it is necessary to be aware of the approach in relation to fire resistance. In other words, the higher the buildings are, the greater the safety requirements must be and, consequently, the covering of its structures.

Next, the case study used as a pilot project will be presented, and the application of the proposed system, previously described, will be discussed.

## 5. Case Study

Here, a case study is presented with the aim to essay the application of the integrated retrofit strategy based on timber pre-assembled components added to building envelopes. The selected building is part of a social housing complex located in the north of Portugal, more precisely in Viana do Castelo, characterized by its proximity to the Lima River (about 500 m) and by the convergence of two important infrastructures: National Road 13 and the railway line (Figure 6).

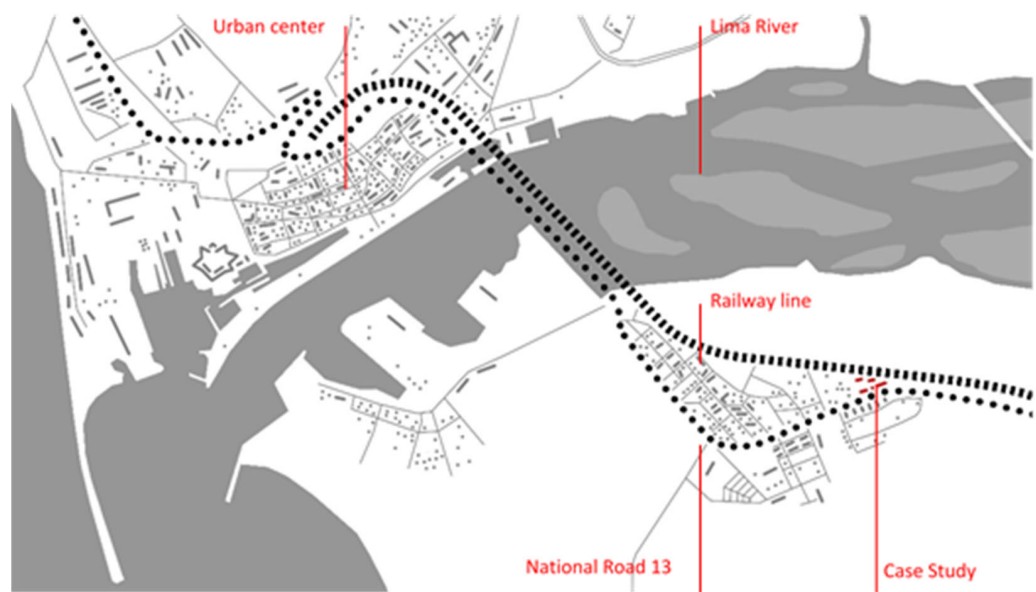

**Figure 6.** Urban context and location of the case study.

This housing complex is composed of four blocks. The block studied (block 4) was obtained through the conjugation of two distinct modules (Figure 7). Module A has a regular rectangular floor plan with four floors, which corresponds to two duplex typologies, and the access is made through exterior galleries; module B has an irregular plan with three floors. Although this case study was built in the early 1980s, it has several anomalies and low quality of construction. In addition to its poor state of conservation, especially on the north façades, the rehabilitation project for blocks 1 and 2 through the application of the ETICS system, which is currently being finalized, made it possible to compare this system with the proposal here presented.

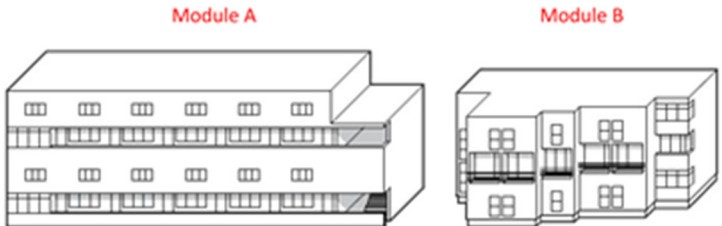

**Figure 7.** Modules that make up block 4.

### 5.1. Existing Building's Structure

From the façade, it is possible to understand the structural mesh of the building, as well as the distribution of housing typologies. Module A is composed of a structural frame of about 4.9 m, and the structural frame of module B varies between 6.10 and 2.85 m (Figure 8).

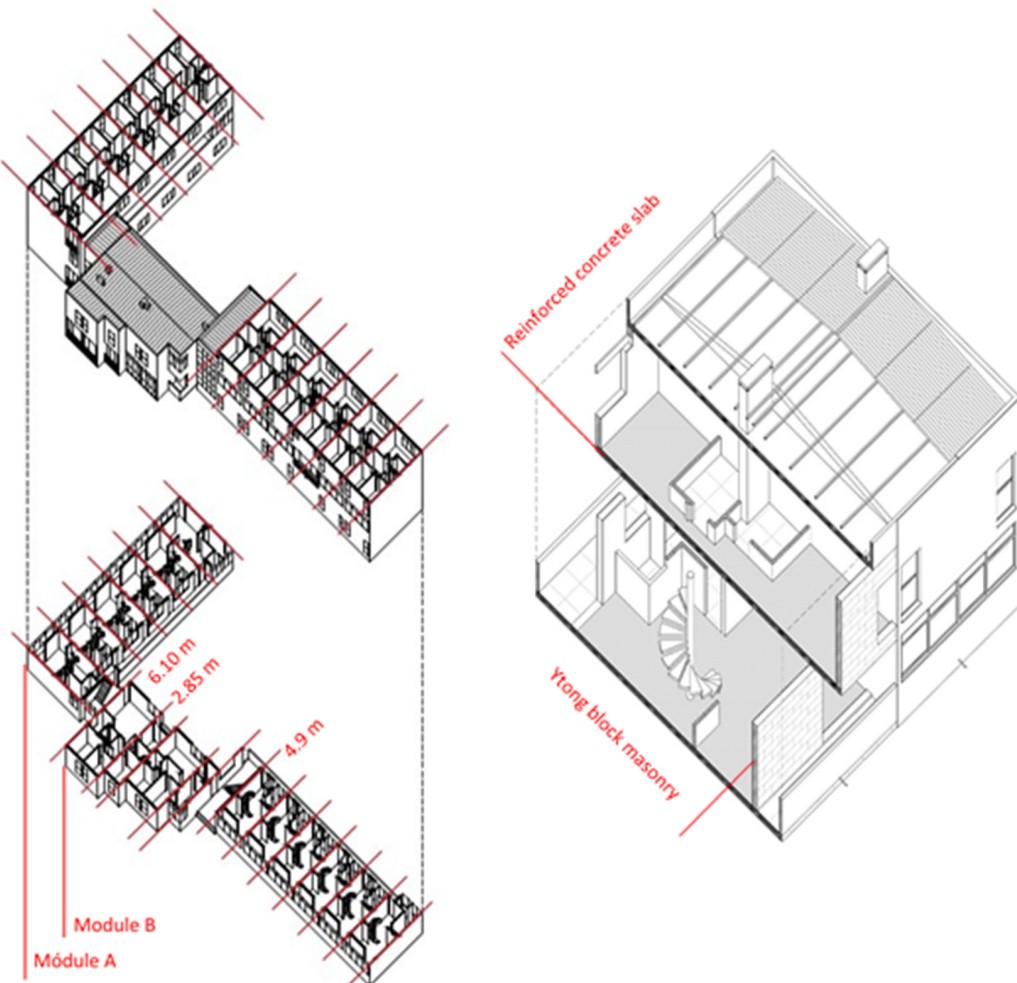

**Figure 8.** Representation of the structural frame (**left**) and the structure elements (**right**).

The main structure of both buildings, modules A and B, is made of reinforced concrete columns and beams with masonry infills. The slabs are also in reinforced concrete. All filling walls, both exterior and interior ones, are built with Ytong autoclaved aerated concrete blocks, plastered and painted with plastic paint. Exterior walls have a thickness of 20 cm while the interior ones are 10 cm thick.

Ytong blocks are obtained by mixing several elements, namely, Portland cement, silica sand, aluminum powder and water. Subsequently, this mixture is placed in molds with pre-placed reinforcement bars and cut to the desired dimensions. Furthermore, it is placed in an autoclave where it will be hardened. One of the characteristics of aerated concrete is its ability to insulate against cold and heat, allowing the construction of simple and solid walls without the need for additional insulation. This is why, in the context of the know-how and experience of the 1980s when the building was built, no additional thermal insulation was applied in this selected case study. However, it should be noted that the thickness of the Ytong blocks used on the exterior walls is not adequate for the current thermal needs and codes.

The current conservation state of the buildings is mainly the consequence of the proximity of the Lima River and the sea. A homogeneous crack pattern of the outer

coating is visible, with some local signs of corroded steel rebars with concrete spalling. The northern façades are the ones more degraded, with a very evident presence of mold and efflorescence.

*5.2. Proposed Intervention*

Any intervention on the selected case study must solve the thermal inefficiency of the building. Moreover, the intervention shall improve the appearance of the building, degraded by the aging of the materials and by the presence of different degradation agents. On the other hand, since the confinement derived from the pandemic caused by the SARS-CoV-2 virus, it became important to increase the interior living space to accommodate the current needs of the families.

In this context, it is proposed to use a prefabricated wood-based system capable of acting in three distinct areas: energy efficiency, structural safety and spatial organization. As an example, it is proposed to intervene in the exterior of block 4 by placing an exoskeleton, turning it into a habitable envelope (Figures 9 and 10).

In terms of energy, the characteristics of the proposed prefabricated system make it possible to increase thermal and acoustic comfort through the placement of insulation in the surroundings and the installation of high performing windows, with double glazing and a sealing system; improve indoor air quality, with the addition of outdoor spaces (balconies) that allow for indoor ventilation; and create intermediate thermal buffer zones (winter gardens), which allow the interior temperature of the dwellings to be controlled throughout the year. The structural resilience is guaranteed through the structure of the proposed system, which is fixed to the reinforced concrete structure, ensuring greater strength and rigidity. Thus, it allows the distribution and reduction of lateral loads and the increase in structural resilience. Regarding the spatial organization of each apartment, we worked in interior spaces and exterior spaces. Private interior spaces are characterized by their versatility and ability to adapt to the needs of the occupants. In the bedrooms, spaces have been created that can function as an office for teleworking or videoconferencing, a reading area, or even a closet (giving more space to the rest of the room). In the social areas, it was decided to add a winter garden. In addition to being able to be used for leisure, socializing and cultivation of plant species, it also works as a thermal buffer zone. It is worth mentioning the cooperation between the winter garden and the balcony, since these spaces can obtain different configurations throughout the year, according to the functional and thermal needs. In winter, the window frames are closed, creating a thermal buffer zone and protecting the interior spaces from low outside temperatures, while in summer, the window frames are open, in order to increase air circulation and prevent overheating of the winter garden. With this, it is possible to merge the balcony and the winter garden and create an outdoor space with greater dimensions. The modules added to the existing building can interact with the outside through windows and doors. Its position and dimensioning are flexible and can be adjusted to each building, depending on ventilation and lighting requirements. Moreover, the windows and doors are not fixed and can be adjusted to the needs of each resident and even to the regulations of different countries in which the system is being applied.

Whenever possible, we created private outdoor spaces as it allows occupants to enjoy the outside air without having to leave their homes. Such spaces are found on the balconies present in all apartments and in the courtyard on the lower floor of module A. The balconies function as an extension of the interior space, where it is possible to develop various activities related to leisure and rest or even dry clothes naturally. In addition, it allows ventilation of the interior of the building, improving the indoor air quality, and it plays an important role if there is a need for the occupants to remain in prophylactic isolation. On the lower floor of the exterior access galleries, it was decided to create a courtyard entrance for each dwelling, ensuring privacy for residents. This covered courtyard, in addition to allowing direct access to the dwellings, also functions as a transition space between the exterior and interior of the house.

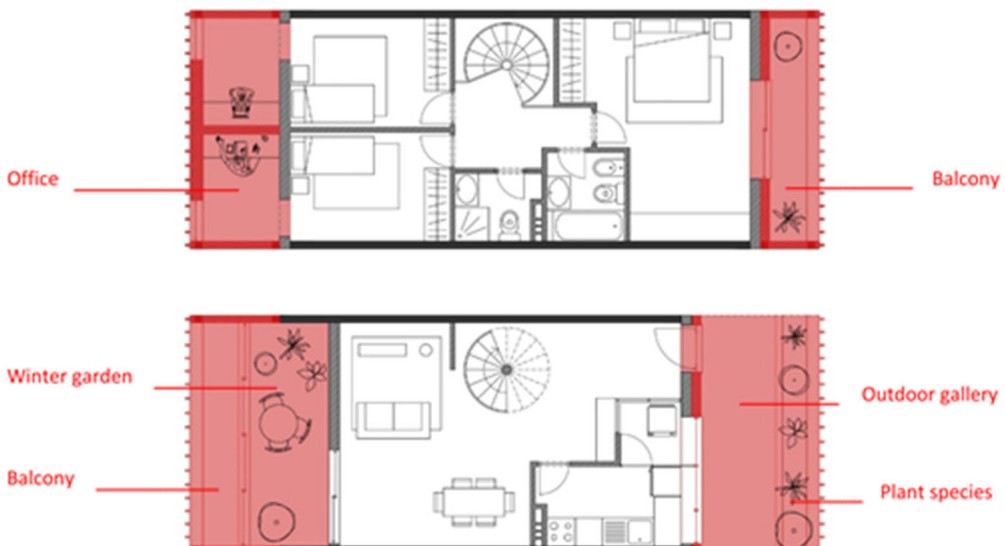

**Figure 9.** Plan of the added spaces.

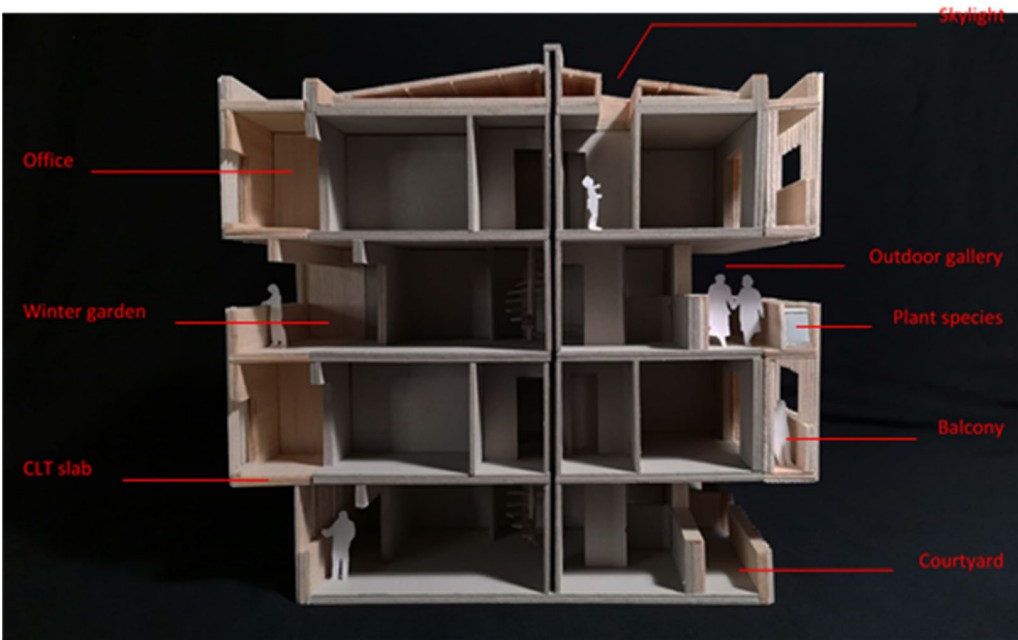

**Figure 10.** Model of the proposal section.

The public spaces created are the exterior galleries and the roof of module B. On the upper floor, the objective was to make the exterior gallery an extension of the surrounding urban space by enlarging it and placing plant species. This allows residents and local people to enjoy the spaces created and enables the creation of a community environment. In module B, we decided to make the roof accessible, in order to consolidate the interaction between people and take advantage of its location to create a viewpoint over the Lima River. This space works as an open garden through the placement of green spaces for the cultivation of small plants (Figure 11).

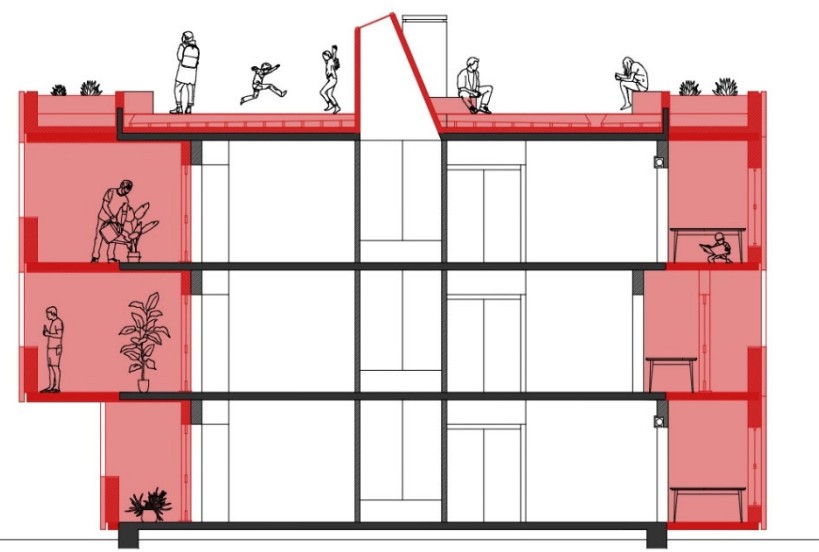

**Figure 11.** Section of module B.

In addition, the placement of skylights makes it possible to light and ventilate the interior spaces, such as bathrooms and vertical accesses (spiral stairs). Unlike module B, the roof of module A is not accessible; however, it was decided to introduce skylights and photovoltaic panels. The façade is composed of wooden slats in which the distance between vertical elements varies according to the spaces. On the ground floor, a smaller distance is used to ensure greater privacy for the occupants. Moreover, it allows controlling the incidence of sunlight. In outdoor spaces, such as balconies and outdoor galleries, the wooden slat is interrupted occasionally, in order to represent the geometry of the original window frames while maintaining the design characteristics. Over time, the vegetation located on the roof and in the outer gallery may extend through the wooden slats, creating a green façade (Figure 12).

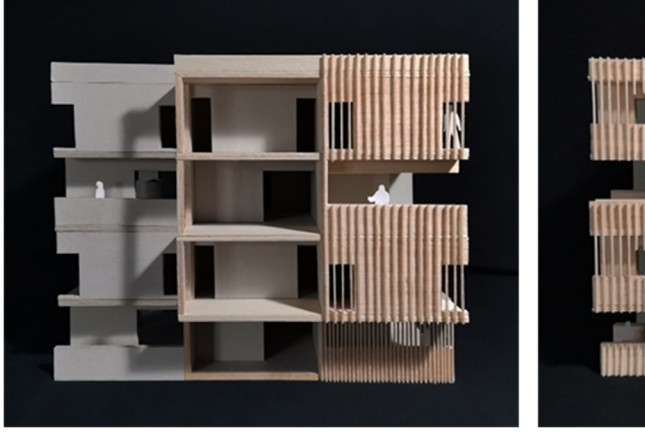 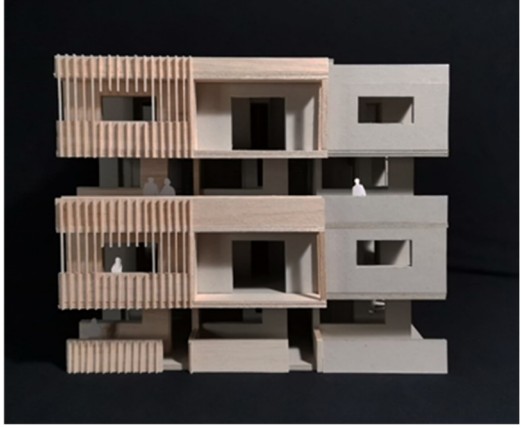

**Figure 12.** Model of the north elevation (**left**) and south elevation (**right**). Representation of three stages: current building, structure and final result.

*5.3. Materials*

The proposed intervention is composed of glulam elements for the columns and CLT panels for the slabs in order to allow continuity of spaces and because it is possible to adapt to any construction due to the factory cut of the existing building's geometry. Moreover, the interior cladding is made of OSB panels, and the exterior cladding is composed of a wooden slat with a surface protection treatment. The choice of wooden slats as an exterior covering is based on its versatility, since, depending on the distance between the vertical

elements, they guarantee greater privacy for residents and greater control of sunlight. The wooden substructure, with vertical and horizontal slats, not only supports the exterior cladding but also allows for the creation of a ventilated façade.

The choice of these materials is due to the fact that they have a high degree of prefabrication. Therefore, the normalization and standardization of the dimensions of these elements and their production in the factory reduces the construction and assembly time and reduces the life cycle cost. Furthermore, as the composition of these materials is based on wood, it makes this solution, and, consequently, the case study building, more efficient and sustainable, since wood is characterized by its ability to store carbon dioxide and for being a natural, renewable and ecological material.

The introduction of mineral wool insulation not only allows for greater thermal and acoustic comfort but also guarantees fire resistance. Moreover, the charring approach also contributes to greater fire safety, as the outer layer of wood, when exposed to fire, reaches its point of combustion. In this chemical reaction, heat removes hydrogen and oxygen from the solid wood, leaving a layer of coal, composed mainly of carbon. This outer layer has a low conductivity, reducing fire progression. In addition to the coal layer, a pyrolysis zone is also formed, where the decomposition of wood occurs due to the increase in temperature of the outer layer. The inner core of the wood retains its mechanical properties, and only its moisture content is reduced.

## 6. Conclusions

The current use of ETICS in Portugal to retrofit the existing buildings stock should be questioned since there are other energy-efficient solutions that also improve the structural stability and increase the architectural aesthetics and organization, with the significant advantage of adding new spaces and rooms adequate to the confinement derived from the present pandemic caused by the SARS-CoV-2 virus. The use of prefabricated wood-based elements and/or systems has demonstrated to increase the energy efficiency, interior air quality, acoustic comfort, waterproofing, aesthetics and real estate value of the buildings, through the protection and placement of insulation in the envelope and the redesign of the façades. Moreover, the rationalization and optimization of the modular constructive elements will reduce the renovation operation time and increase the speed of manufacturing and assembly. As defended by [29], there is still a long way for building scientists and professionals to go in order to make existing building stock more energy efficient and environmentally sustainable.

A holistic and integrated intervention based in the use of prefabricated wooden systems attached to the existing structure was presented and applied to a selected reinforced concrete building from the 1980s. It was possible to improve the energy efficiency of the building while adding volumes and living spaces. More studies are certainly needed but the essay was promising.

**Author Contributions:** C.M. (Cláudio Meireis) and F.S.S., methodology, formal analysis, investigation, writing—original draft preparation; C.M. (Carlos Maia), A.C.F. and J.M.B., conceptualization, methodology, formal analysis, writing—review and editing, supervision. All authors have read and agreed to the published version of the manuscript.

**Funding:** This research received no external funding.

**Informed Consent Statement:** Not applicable.

**Conflicts of Interest:** The authors declare no conflict of interest.

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
