# Peer review of "Current Practice and Potential Associated with Timber-Based Solutions for Buildings Retrofitting"

_infrastructures, doi:10.3390/infrastructures7020025_

Round 1

Reviewer 1 Report

The manuscript can be accepted

Author Response

The authors would like to sincerely thank the reviewer for the comments, which certainly increased the quality of the manuscript by being considered in the revised version. The authors hope that the new version of the paper can be accepted for publication.

Reviewer 2 Report

The paper deals with a possible strategy of energetic, structural and architectural improvement of existing buildings by the means of timber-structures exoskeletons. The theme is of sure interest, however it is the reviewer's opinion that the methodological approach of the paper should be improved as follows. An overall description of the proposed system is needed, before any application to a case study, in order to understand its replicablity to other cases. A discussion of the proposed solution in respect to technical aspects (e.g. connections to the r.c. frame or foundations) is also recommended.

The authors will find specific comments to their text in the attached pdf file.

Author Response

The authors would like to sincerely thank the reviewer for the comments, which certainly increased the quality of the manuscript by being considered in the revised version. The authors hope that the new version of the paper can be accepted for publication.

Comments to the Author:

The paper deals with a possible strategy of energetic, structural and architectural improvement of existing buildings by the means of timber-structures exoskeletons. The theme is of sure interest, however it is the reviewer's opinion that the methodological approach of the paper should be improved as follows.

- An overall description of the proposed system is needed, before any application to a case study, in order to understand its replicablity to other cases. A discussion of the proposed solution in respect to technical aspects (e.g. connections to the r.c. frame or foundations) is also recommended.

Reply: The authors agree that the description of the proposed system is not clear and does not reveal its characteristics that allow it to be replicated in other buildings. For that, the materials that compose the proposed system and the function they perform were explained, both in text and in an axonometric that represents all the layers. Moreover, the choice of structure elements and the absence of foundations were discussed.  

“The structure of the prefabricated system is composed of glulam columns and beams and CLT slabs. The columns allow the continuity of spaces, and, since there are buildings with an irregular exterior envelope, generally caused by balconies, the use of CLT panels for the slabs will allow it to adapt to any kind and geometry of construction. In fact, in factory, and using the available digital technology of drawing and CNC cuts, it is possible to have a geometry of the CLT panel fitted to the existing building. As wood is a light material (strength/weight ration), its use in this system allows the absence of additional foundations, with the prefabricated modules being directly supported by the reinforced concrete structure of the existing building.

In Figure 5 the components of the proposed system are represented in axonometry through the decomposition of elements. In addition to identifying the layer and their functions, it is also possible to perceive the relationship between the different elements.

 Figure 5. Structural and functional layers of the proposed prefabricated system.

Regarding thermal insulation, mineral wool was chosen to guarantee grater safety against fire, and greater thermal and acoustic comfort. As wood has a low thermal conductivity, the insulation is placed in the interstices of the structural layer. Thermal bridges are minimized through the thickness of the thermal insulation and the placement of a wood fiberboard, which acts as additional insulation.

The interior cladding is made of OSB (Oriented Strand Board) panels and the exterior cladding is composed of wooden slat with a surface treatment and protection. The choice of wooden slats as an exterior coating is based on its versatility, since, depending on the distance between the vertical elements, it guarantees privacy for occupants and it controls the sunlight. The wooden substructure, with vertical and horizontal slats, not only sup-ports the exterior cladding, but also allows for the creation of a ventilated façade.

Although there are various functions that the building envelope must fulfil, there are layers that can perform several functions at the same time. In this regard, the structure functions as a structural layer and interior cladding; the OSB panel has the function of interior cladding, but also performs the function of airtightness through the placement of single-sided adhesive tapes, and a vapor barrier; the wood fiberboard is pressure resistance and vapor permeable, and has the function of additional insulation, protection of structural components, and also a second layer resistant to water and wind.

One of the properties of wood is its combustibility. This is one of the main reasons why people, in general, do not feel safe in buildings with wooden structures, particularly tall buildings [21]. Although wood is a combustible material, there are two distinct approach that allow a sufficient fire performance of a solid timber structural system: char-ring and encapsulation. Charring is a process in which, when exposed to fire, the outer layer of wood reaches its point of combustion, creating a layer of charcoal. This outer layer has a low conductivity, reducing fire progression. The inner core of the wood retains its mechanical properties, and only its moisture content is reduced. Therefore, charring is an approach that allows exposing the solid timber structure outside the building. While encapsulation is an approach where the system is coated with fire resistant materials such as plasterboard panels. In this regard, depending on the height of the buildings in which it will be intervened, it is necessary to be aware to the approach in relation to fire resistance. In other words, the higher the buildings are, the greater the safety requirements must be and, consequently, the covering of its structures.

Next, the Case Study used as pilot project, will be presented and the application of the proposed system, previously described, will be discussed.”

- Please enrich the discussion of the downsides of these systems.

Reply: The authors recognize that the disadvantages of the Prefabricated Cladding Solutions are not explicit, therefore, the downsides of these systems were introduced and complemented in the revised manuscript, as shown below:

“Although these systems allow multifunctional interventions and adaptability to different climate zones, they don’t take full advantage of the existing buildings, such as spatial organization. In other words, it does not qualify the pre-existing interior spaces, maintaining a static and rigid organization.”

- Are these additions compliant with the Portuguese building codes in terms of natural ventilation and lighting requirements? Probably changes to windows and doors are needed. Please, discuss.

Reply: As the windows and doors are not fixed and can be adjusted to the needs of the residents, there should be no problems with natural ventilation and lighting requirements. However, each country's building code must be verified to fulfil all requirements.

“The modules added to the existing building can interact with the outside through windows and doors. Its position and dimensioning are flexible and can be adjusted to each building, depending on ventilation and lighting requirements. Moreover, the windows and doors are not fixed and can be adjusted to the needs of each resident and even to the regulations of different countries in which the system is being applied.”

- The difference between the two configurations is not so evident. Please improve the clarity of the figure.

Reply: As the difference between the two images is unclear and does not add much to the rest of the paper, the authors think that the figure in not relevant and have decided to eliminate it.